# Effects of Native Whey Protein and Carbohydrate Supplement on Physical Performance and Plasma Markers of Muscle Damage and Inflammation during a Simulated Rugby Sevens Tournament: A Double-Blind, Randomized, Placebo-Controlled, Crossover Study

**DOI:** 10.3390/nu14224780

**Published:** 2022-11-11

**Authors:** Marina Fabre, Bertrand Mathieu, Eve Tiollier, Cédric Leduc, Matthieu Clauss, Alexandre Marchand, Julien Robineau, Julien Piscione, Tanguy Serenari, Jacqueline Brasy, Mathilde Guerville, Amandine Ligneul, Xavier Bigard

**Affiliations:** 1Laboratory Sport, Expertise and Performance (SEP, EA 7370), French Institute of Sport (INSEP), 75012 Paris, France; 2French Rugby Federation, 91460 Marcoussis, France; 3Carnegie Applied Rugby Research (CARR) Center, Institute for Sport, Physical Activity and Leisure, Carnegie School of Sport, Leeds Beckett University, Leeds LS1 3HE, UK; 4Sport Science and Medicine Department, Crystal Palace FC, London SE25 6PU, UK; 5French Anti-Doping Agency (AFLD), 75009 Paris, France; 6Nutrition Department Lactalis Recherche et Développement, 35134 Retiers, France; 7Union Cycliste Internationale (UCI), 121860 Aigle, Switzerland

**Keywords:** recovery, rugby, performance, cytokines, microRNAs

## Abstract

The importance of optimized recovery during a sport competition is undisputed. The objective of this study was to determine the effects of recovery drinks comprising either carbohydrate only, or a mix of native whey proteins and carbohydrate to maintain physical performance and minimize muscle damage during a simulated rugby sevens (rugby 7s) tournament. Twelve well-trained male rugby players participated in three simulated rugby 7s tournament days with a week’s interval in between. Each tournament comprised a sequence of three simulated matches, interspersed with 2 h of recovery. Three different recovery drinks were tested: a placebo (PLA, nonenergetic chocolate-flavored drink), a carbohydrate drink (CHO, 80 g of carbohydrate) or an isoenergetic carbohydrate–protein drink (P-CHO, 20 g of Pronativ^®^, native whey protein and 60 g of carbohydrate). A different recovery drink, consumed after each match, was tested during each simulated tournament. Physical performance, muscle damage and muscle pain were assessed before and after each simulated tournament. Regarding physical performance, both P-CHO and CHO drinks had a positive effect on the maintenance of 50 m sprint time compared to the PLA drink (effect sizes *large* and *moderate*, respectively). Regarding muscle damage, the P-CHO supplement attenuated the creatine phosphokinase increase at POST6 compared to PLA (effect size, *moderate*). Finally, P-CHO and CHO drinks reduced the exercise-induced DOMS (effect size, *moderate*), compared to the PLA condition (effect size, *large*), while P-CHO only reduced pain on muscle palpation and pain when descending stairs compared to PLA 24 h post-tournament (effect size, *small*). This study suggests that consuming a recovery drink containing native whey proteins and carbohydrate or carbohydrate only after each match of a rugby 7s tournament may attenuate the exercise-induced increase in markers of muscle damage and maintain physical performance.

## 1. Introduction

Rugby sevens (rugby 7s) is a high-intensity intermittent team sport. Competition is played during a two-day tournament, with several matches each day. Generally, players compete daily during two to three matches, for two half-times of seven minutes with two minutes of recovery between each half with 1 h 30 to 3 h of rest between matches. This Olympic sport is characterized by physiological constraints associated with several repetitions of maximal running sprints with short times of recovery, and muscular constraints induced by the repetition of several accelerations, decelerations, braking and tackles [1,2]. Consequently, rugby 7s competitions may induce [1] glycogen depletion and [2] muscular damages that may impair performance. Thus, optimized nutrition strategies for recovery after each game are an important performance factor in rugby 7s competitions.

In this context, nutritional strategies aimed to restore energy and fluid deficits and to repair muscle damage at the end of every match may be a determinant factor for enhanced performance throughout the tournament. This includes an optimal amount of carbohydrates (for glycogen restoration) [3], liquids and minerals (for rehydration) [4] and proteins, which provide adequate and bioavailable essential amino acids (to induce maximal muscle protein synthesis) [5,6,7].

After resistance exercise, it has previously been shown that native whey proteins together with carbohydrates allow for better muscle recovery compared to casein proteins plus carbohydrates [8] or carbohydrates alone [9]. The consumption of a whey protein and carbohydrate mix after prolonged endurance exercise has been shown to reduce muscle damage [10,11,12,13]. However, very few studies have investigated the impact of a protein and carbohydrate mix on recovery and performance markers after high intensity intermittent exercise [14]. Saunders et al. (2004) observed that supplementation with whey proteins and carbohydrates during and immediately after exercise induced a lower increase in creatine phosphokinase (CPK) compared to carbohydrates alone [11]. Moreover, in orienteering running, the ingestion of whey proteins and carbohydrates before and after each training session induced a lower increase in biomarkers of muscle damage [15]. However, no studies have yet investigated the effect of certain nutrition strategies on match recovery during a rugby 7s competition.

Thus, our objective was to investigate the effects of several nutritional strategies based on native whey protein and/or carbohydrate supplementation on the maintenance of physical performance during a simulated rugby 7s tournament day. We hypothesized that the ingestion of a native whey protein and carbohydrate mix after each match would be associated with lower levels of muscle damage and inflammatory markers, which would help to maintain performance until the end of the day, compared to the ingestion of isoenergetic carbohydrate and nonenergetic placebo beverages.

## 2. Materials and Methods

### 2.1. Subjects

For the purpose of the study, twelve male subelite rugby players from the French rugby federation were recruited. Inclusion criteria were age (between 18 and 30 years old), body weight (between 60 and 85 kg), competition and training level and schedule, passing of a physical preinclusion test and a medical certificate for participation in the study. All the participants had to be covered by the French social security system and licensed with the French rugby federation. Participants were excluded when they followed a medical treatment, took nutritional supplementation treatment, had a history of cardiovascular or any other inflammatory diseases, colds or flu, acute respiratory infection, dental problems, renal/hepatic abnormalities, lactose and/or milk protein intolerance, followed a vegetarian diet and had a recent history of muscle injury.

### 2.2. Experimental Design

This study was carried out in agreement with the guidelines set by the Declaration of Helsinki of 1975, as revised in 2013, and was approved by the local Ethics Committee (Sud-Est VI Clermont-Ferrand, France, ref AU 1323/2017-A00653-50).

It was a double-blind placebo-controlled, randomized and crossover interventional study. Each player participated in three simulated rugby 7s tournament days with a week’s interval in between. Two months preceding the start of the study, all participants had to stop taking any nutritional supplements, including proteins. Subjects were made familiar with all the different physical testing requirements during a preparatory period, before the start of the study. The study took place at the Research Department of the French National Institute of Sport (INSEP, 11 avenue du Tremblay 75012 Paris France) between 28 April and 22 May. Subjects followed the same training protocol for 3 consecutive weekends but consumed different drinks every weekend. Beverages were consumed in a double-blind manner after each simulated rugby 7s match (two halves of seven minutes). Participants arrived at the INSEP the night before, where they had a standardized dinner at 21:00. On Saturday morning, a standardized breakfast was provided at 07:00 followed by the first blood test (PRE) (Figure 1). At 09:30, physical tests were performed to establish a baseline (PRE test). Then, they performed three simulated rugby 7s matches, with a 2 h resting period between each match. Between matches 2 and 3, a standardized lunch was provided. The experimental beverages were consumed after each match. The simulated rugby 7s tournament day ended with a POST test session (POST test). In addition to the tests performed in PRE, an intense intermittent validated test (Yo-Yo IR2 test, [16]) was performed at POST.

Blood samples were collected at four different time points: PRE, just after the end of the post-tests (POST), 6 h after the tests (POST6) and 12 h after the tests (POST12). All participants followed the same protocol for three consecutive weekends. Physical activity, sleep and alcohol consumption were controlled during the week.

### 2.3. Simulated Rugby 7s One-Day Tournament

Because there is great variability of physical performance during each rugby 7s tournament, we used the Rugby 7s Simulation Protocol (R7SP) to examine the effects of the nutritional intervention [17]. This simulation protocol was designed to mimic the locomotor and metabolic demands of a rugby 7s match, with the distance per match being 1480 m, the average distance per minute 105.7 m and the time spent sprinting (>6 m/s) 12.1%, high-intensity running (5–6 m/s) 4.8%, striding (3.5–5 m/s) 19.1%, jogging/cruising (2–3.5 m/s) 21.4%, standing/walking (0–2 m/s) 33.1% and grappling (0 m/s) 9.5% [2,17].

### 2.4. Nutritional Strategies

The supplements tested in the study were presented in the form of a chocolate-flavored drink (powder mixed with water). Supplements were manufactured by JLB Development (18 chemin des Tard Venus–69530 Brignais–France). Three different supplements were tested:-A protein-carbohydrate drink (P-CHO), delivering 20 g of Pronativ^®^ native whey proteins (Pronativ^®^ native whey proteins, Lactalis Ingredients, Bourgbarré, France), 60 g of carbohydrates (50/50; glucose/maltodextrin) and 0.5 g of fat.-An isoenergetic carbohydrate drink (CHO), delivering 80 g of carbohydrates (50/50; glucose/maltodextrin) and 0.5 g of fat.-A nonenergetic placebo drink (PLA) with the same chocolate flavor.

Subjects consumed two 500 mL beverages to provide a carbohydrate concentration of less than 8%. The first one was drunk within 15 min after the end of simulated rugby 7s matches and provided 20 g of proteins and 20 g of carbohydrates (P-CHO) or 40 g of carbohydrates (CHO). The second 500 mL beverage provided 40 g of carbohydrates for both the P-CHO and CHO scenarios and had to be drunk within the 45 min after the end of the matches.

### 2.5. Randomization, Blinding and Compliance

The supplements were packaged in individual unidose sachets and numbered with the study subject’s randomization number. To ensure a random distribution and to limit bias, the randomization list was prepared by a researcher associated with the study, but who was not present during the test sessions, including the inclusion tests. In addition, the researchers who prepared the nutritional supplements did not perform the performance tests and those who did were not made aware of which drinks were given at which timepoints. The researchers who prepared the study drinks also managed the timepoints at which the drinks were ingested.

### 2.6. Diet Control

Precise meal composition recommendations were provided to the players for the 2 days preceding each rugby 7s simulated tournament, in order to cover the energy and macronutrient requirements of highly trained players [18]. For this purpose, standard menus were provided. Moreover, the dinner the day before the simulated tournament was standardized, as explained below. The purpose was to normalize the macronutrients and especially the carbohydrates before each tournament.

For the day of the simulated tournament, the buffet options were standardized for all meals (breakfast, lunch and dinner). Breakfast options included eggs or ham, bread, fruit or fruit juice and, if preferred, yogurt, milk, cheese, butter, jam, honey or chocolate powder. Lunch was a rice salad with chicken, a yoghurt and a cereal bar. Dinner was a mixed salad, grilled meat or fish with pasta or rice, a yoghurt and/or some fruit. During the first weekend, participants could choose from the buffet, what and how much they wanted. Food intake was noted, both quantitatively and qualitatively. For the following weekends, the same meals were prepared for all participants to ensure that they consumed the same food throughout the three testing weekends. Energy, protein, carbohydrate and lipid intakes, as well as some micronutrients, during the follow-up periods were calculated for each meal using a dietary analysis based on information from the French database of food composition sources (CIQUAL, 2013) (Table 1).

### 2.7. Performance Analysis

We chose to evaluate general physical performance, which is required during a rugby 7s match, such as running at maximal speed (sprints from 10 to 50 m), lower body maximal power output (countermovement jump, CMJ) and ability to perform intermittent exercise (Yo-Yo IR2 test).

Speed. Sprint performance was assessed during two 50 m bouts, using a wireless sports timing system displaying every 10 m (Smart Speed, Fusion Sport, Australia) with a 0.01 s accuracy. Players started each sprint from a standing position with their feet set 0.5 m behind the first timing gate. Each trial was followed by 4 min of recovery. The mean of the two trials were used in the analysis to ensure better reliability.

Maximal power output. Power performance was measured using a portable linear encoder device (GymAware Power Tool, kinetic Performance Technologies, Canberra, Australia). Jump assessments required each participant to perform 4 CMJs with a broomstick placed behind the neck to limit swing peak velocity (m/s). Mean power (W) and jump height (cm) were used to quantify neuromuscular performance. The mean of the trials (excluding the best and the worst value) was calculated [19].

Ability to perform repeated intense exercises. All participants completed the Yo-Yo intermittent recovery level 2 test [20]. The Yo-Yo IR2 is a maximal aerobic capacity test consisting in repeated 2 × 20 m shuttle runs that are performed at progressively increasing speeds. These workloads were interspersed with 10 s active rest periods in which participants were instructed to walk around a cone 5 m away. This recovery period differentiates the Yo-Yo IR2 from other multistage fitness tests, making it more specific to intermittent athletic football movement patterns [21]. Participants were instructed to run in time with the sound signal that occurred at shorter and shorter intervals as the test progressed. The test was over when a participant could no longer reach the cone at the time of the audible signal (twice in a row, or voluntarily terminated). Yo-Yo IR2 is an ecologically valid indicator of the capacity to perform intense intermittent exercise with a large anaerobic component associated with a significant aerobic contribution [16].

Specific performance. At the end of each half-time, the participants conducted a specific rugby test that consisted of a circuit training with tackles, sprints and active recovery to be done as quickly as possible [17]. The time reached to complete this task was used. The performance realized during the first game was used to assess the reliability of this performance test.

### 2.8. Physiological Load Quantification

During every rugby 7s simulation session, participants wore a Global Positioning System (GPS) unit capturing data at 16 Hz (SensorEverywhere, Digital Simulation, Paris, France). GPS units were positioned in a customized pocket placed in their shirt and located between the scapulae. To limit a potential interunit variability, each player wore the same unit for the total duration of the study. The GPS data were captured and computed with SensorEverywhere software (Digital Simulation, France).

### 2.9. Blood Analysis

A nurse collected 2 blood samples (PRE, POST, POST6, POST12) on each test day (in the antecubital fossa vein). Blood samples were collected into 5 mL lithium heparin blood-collection tubes for cytokine analyses and in EDTA tubes (Greiner Bio-one, Frickenhausen, Germany) for microRNA analyses. These tubes were centrifuged at 4500 rpm for 10 min at 4 °C within 10 min after collection. The plasma supernatant was separated, and samples were stored in 2 mL Eppendorf tubes at −80 °C for further analysis.

Cytokines. Cytokines, including inflammatory and anti-inflammatory biomarkers (IL-6, TNF-alpha, IL-1ra), were measured using ELISA kits. These analyses were done by the Immunology Laboratory of the Hospital Center Lyon Sud (France). Quality controls were also performed for all analyses.

microRNAs. The response to exercise of three microRNAs (miR-1, miR-146a, miR-208a) were assessed. These microRNAs were based on their tissue-specific (muscular for miR-1, cardiac for miR-208a) or context-specific (systemic inflammation for miR-146a) expression [19,22,23]. In addition, miR-93 and miR-191 were chosen as controls for normal expression [24,25]. Total RNAs including miRNAs were extracted from 200 µL plasma using the miRNeasy Serum/Plasma Kit (Qiagen, Courtaboeuf, France) and eluted in 18 µL of RNAs-free water, according to manufacturer instructions. RNAs were stored at −80 °C. cDNA was obtained from 2 µL total RNA using the TaqMan^®^ Advanced miRNA cDNA Synthesis Kit (Thermo Fisher Scientific, Illkirch, France). Specific microRNA quantifications were then performed using Taqman Advanced miRNA assays (Thermo Fisher Scientific). miRNA expression was measured in triplicate using TaqMan^®^ Fast Advanced Master Mix (Thermo Fisher Scientific) and each Taqman Advanced miRNA probe on a CFX96 Touch™ Real-Time PCR Detection System (Bio-Rad, Marnes-la-coquette, France). A miRNA was considered expressed if the mean Cq value obtained was under 36, and higher values were not considered reliable. For each target miRNA (miR-1, miR-208a and miR-146a), the relative changes of expression compared to the basal value (PRE) were calculated using the 2^−ΔΔCq^ method [26] and using the geometric mean of the Cq values obtained for each endogenous control (miR-93 and miR-191) as the control Cq value [26].

Creatine phosphokinase (CPK). Blood CPK concentrations were measured using approximately 500 μL of blood collected from fingertip capillary pricks and stored in a microtube containing lithium heparinate (BD Microtainer, BD, Franklin Lakes, NJ, USA). Within one hour after blood collection, 32 μL was taken from the tube using a specific pipette and placed on a measurement strip. Analyses were performed using a Reflotron Sprint (Roche Diagnostics, Grenzacherstrasse, Switzerland). The Reflotron was calibrated according to manufacturer recommendations. Blood for CPH analyses was collected before the first match of the day (PRE, between 7 and 8:30 am) and at the end of the day after all matches (POST, POST6 and POST12).

### 2.10. Statistical Analysis

The statistical analyses were performed in several steps. The Gaussian distribution of the data was assessed using the Shapiro–Wilk normality test. If the variables were distributed normally, changes overtime were tested using a one-way (nutritional strategies) ANOVA or two-way ANOVA (nutritional strategies × time). Tukey’s post hoc analysis was performed when a significant F ratio was shown to isolate specific differences. For data that did not pass the normality test, the Friedman test was used. If this was significant, pairwise comparisons were done using Durbin’s test. Statistical significance was defined as *p* < 0.05. Data are presented as mean ± SD.

In order to assess the repeated bout effect, the Friedman test was used to compare the physiological performance across the three simulated rugby 7s tournaments.

In addition, data were analyzed using the magnitude-based inference approach [27]. This qualitative approach was used to analyze performance and biologic parameters because traditional statistical approaches often do not indicate the magnitude of an effect, which is more relevant than any statistically significant effect to infer practical recommendations for athletes. To reduce any possible bias arising from a nonuniformity of the error, all data were log-transformed before the analyses. The magnitudes of the within-group changes were interpreted by using effect size (Cohen’s d) values and classified as small (ES: 0.2–0.6), moderate (ES: 0.6–1.2), large (1.2–2.0), and very large (2.0–4.0). Quantitative chances of higher or lower values than the smallest worthwhile change (SWC, equal to a Cohen’s d of 0.2) were evaluated qualitatively as follows: <1%, almost certainly not; 1–5%, very unlikely; 5–25%, unlikely; 25–75%, possible; 75–95%, likely; 95–99%, very likely; and >99%, almost certain. When the beneficial or detrimental effects were both >5% higher or lower than 5%, the true difference was assessed as unclear [27].

## 3. Results

### 3.1. Subjects and Dietary Intake

Twelve well-trained subelite male rugby players (age 22.9 ± 5.3 years; height 179 ± 10.9 cm; body weight 80.4 ± 12.7 kg) participated in the study.

Dietary intake during the rugby 7s tournaments is reported Table 1. Daily dietary intake without supplements (PLA condition) were 2656 ± 559 kcal of total energy, including 1.9 ± 0.4 g/kg of proteins, 4.2 ± 1.1 g/kg of carbohydrates and 0.9 ± 0.3 g/kg of lipids. When experimental beverages were consumed, daily energy intake was 3616 ± 512 kcal. Daily protein intake was 1.9 ± 0.4 g/kg and 2.6 ± 0.4 g/kg with the CHO and P-CHO beverages, respectively. Daily carbohydrate intake was 7.1 ± 1.2 g/kg and 6.3 ± 1.1 g/kg with the CHO and P-CHO beverages, respectively. Energy intake was higher in CHO and P-CHO compared to PLA (*p* < 0.001, *large* ES: 2). There was no significant difference in energy intake between the CHO and P-CHO conditions. As expected, protein intake was higher in the P-CHO than in the CHO and PLA conditions (*p* < 0.001, *large* ES: 1.6), without any difference between the CHO and PLA conditions. Carbohydrate intake was higher in the CHO and P-CHO than in the PLA condition (*p* < 0.001, *very large* ES: 2). Moreover, carbohydrate intake was *most likely* higher in the CHO than in the P-CHO condition (*small* ES: 0.5).

### 3.2. Physical Performances

The over-time variability of the total distance covered during each match as measured with GPS was examined with a Friedman test on data collected during the total of nine matches (three tournaments of three matches). No significant variation in the distance covered was found. Therefore, the workload and physiological demand could be considered to have remained constant throughout the study.

Moreover, no order effect was detected on physical performance markers and no cumulative effect of tournament repetition on physical performance was detected.

#### 3.2.1. Mean Power Output

Mean power output during CMJ was not affected by the repeated simulated rugby 7s matches nor by the nutrition supplementation strategy. For PLA, the mean power was 3272 ± 454 W at PRE and 3403 ± 461 W at POST (an increase of 131 W or 4%, unclear). For CHO and P-CHO, the mean power was, respectively, 3206 ± 493 W and 3293 ± 593 W at PRE and 3272 ± 454 W and 3352 ± 620 W at POST (an increase of 59 W or 1.8%, likely trivial for both conditions).

#### 3.2.2. High-Intensity Performance

The quantitative analysis did not reveal any statistical difference between the means of the distance covered during the Yo-Yo IR2 test.

Using qualitative analyses, the capacity to perform repeated intense exercises at the end of the simulated tournament *likely* decreased with PLA (distance covered 307 ± 139 m), in comparison with both P-CHO (370 ± 117 m) and CHO (371 ± 134 m) conditions (*small*, ES: 0.5). However, no difference was found between the P-CHO and CHO conditions (Figure 2).

#### 3.2.3. Sprint Performance

Using the Friedman test, there was no statistical effect of the nutritional supplement on the sprint performance.

Using a qualitative data analysis, sprint performance from 10 to 50 m was not altered with both P-CHO and CHO at the end of the tournament day, compared to PRE. However, with PLA, sprint time to achieve a 50 m distance *likely* increased (+1.7 ± 2.1%) (*moderate*, ES: 1.7). Compared to PLA, P-CHO appeared to have a *likely* positive effect on the maintenance of running sprint performance, from the distances of 30 and 40 m (*moderate*, ES: 1.3 and 1.4, respectively), and the magnitude increased at 50 m (*large*, ES: 2.5). Similarly, CHO *likely* contributed to maintain sprint performance compared to PLA but with a lower magnitude effect (*small* at 30 and 40 m, ES: 0.9 and *moderate* at 50 m, ES: 1.8). P-CHO condition tended to have a small positive effect on the 50 m sprint performance compared to CHO (64% chance to get a better performance with P-CHO compared to 6% chance with the CHO beverage). Although this tendency was noticed at 50 m, no differences were found between P-CHO and CHO on sprint performance registered from 10 to 50 m (Figure 3).

### 3.3. Biomarkers of Muscle Damage and Inflammation

#### 3.3.1. Creatine Phosphokinase (CPK)

Using the Friedman test, there was no significant effect of the nutritional supplement on blood CPK measured at PRE and several POST times (Table 2).

Using qualitative data analysis, the one-day simulated rugby 7s tournament *most likely* increased CPK concentration from very large to large, from 1 h to 12 h after the end of the tournament day (POST, POST6 and POST12 compared to PRE), regardless of which beverage was consumed. P-CHO *likely* attenuated the CPK increase at POST6, compared to PLA condition (*moderate*, ES: 0.8). However, there was no difference between P-CHO and CHO, nor between the CHO and PLA conditions.

#### 3.3.2. Tumor Necrosis Factor-Alpha (TNF-Alpha)

The quantitative statistical analysis failed to show a significant effect of the nutritional supplement on TNF-alpha measured at PRE and several POST times (Table 2).

Using a qualitative data analysis, it was shown that the TNF-alpha concentration *most likely* decreased at the end of match day (POST) compared to baseline, without any difference between the conditions (from 1 ± 0.1 to 0.8 ± 0.1 pg/mL for the three conditions, *large*, ES: 1.8 for P-CHO and PLA and ES: 1.7 for CHO). After 6 h of recovery (POST6), the TNF-alpha concentration remained *most likely* lower than baseline values with P-CHO and PLA (*large*, ES: 1.4 and 1.2, respectively), and *very likely* lower with CHO (*moderate*, ES: 0.6). At POST12, the TNF-alpha concentration returned to baseline values with P-CHO and CHO but was *possibly* higher than baseline with the PLA condition (*small*, ES: 0.3).

#### 3.3.3. Interleukin 1 Receptor Antagonist (IL-1ra)

A global effect of the experimental beverages was shown for IL-1ra measured at POST6 (*p* < 0.05). Using the Durbin signed rank test, blood IL-1ra in the PLA condition was higher than in the P-CHO (*p* < 0.01) and CHO (*p* < 0.05) conditions. At POST12, blood IL-1ra in the PLA condition was higher than in the P-CHO condition (*p* < 0.05) (Table 2).

Using a qualitative data analysis, it was shown that after the tournament day (POST), the IL-1ra concentration *most likely* increased compared to baseline, for all conditions (*very large*, ES: 2 for P-CHO and *large*, ES: 1.8 and 1.7 for CHO and PLA, respectively). At POST6 the IL-1ra concentration remained *most likely* elevated compared to baseline for the three conditions (*large*, ES: 1.4 for P-CHO and PLA and ES: 1.3 for CHO). After 12 h of rest, the IL-1ra concentration returned to basal values with P-CHO and CHO but remained *likely* higher than baseline with the PLA condition (*small*, ES: 0.4). At POST6 and POST12, P-CHO *likely* lowered the increase of the IL-1ra concentration compared to the PLA condition (*small*, ES: 0.5). Similarly, at POST6 and POST12, the CHO condition, respectively, *likely* and *possibly* lowered the IL-1ra concentration compared to the PLA condition (*small*, ES: 0.4). However, there was no difference between the P-CHO and CHO conditions.

#### 3.3.4. Interleukin 6 (IL-6)

Using the Friedman test, no global effect of the nutritional supplements was shown on blood IL-6 (Table 2).

After the tournament day (POST), the IL-6 concentration *most likely* increased compared to PRE, for all conditions without any difference (*moderate*, ES: 1). At POST6, the IL-6 concentration remained *likely* higher than baseline values with P-CHO (*small*, ES: 0.4), CHO and PLA (*small*, ES: 0.5), without any difference between the three conditions. After 12 h of rest (POST12), the IL-6 concentration returned to basal values with P-CHO but remained *likely* higher than baseline values with the CHO and PLA conditions (*small*, ES: 0.4). At POST12, the IL-6 concentration was *possibly* lower with P-CHO compared to the CHO and PLA conditions (*small*, ES: 0.2 and 0.3, respectively). However, there was no difference between CHO and PLA.

#### 3.3.5. microRNA (miRNA)

All miRNAs tested were present in the plasma at detectable levels except for miR-208a.

#### 3.3.6. Muscular microRNA (miR-1)

A global effect of the experimental beverages was shown for miR-1 measured at POST (*p* < 0.05). Using the Durbin signed rank test, miR-1 in the PLA condition was higher than in the P-CHO (*p* < 0.01) and CHO (*p* < 0.05) conditions.

MiR-1 expression increased about two- to threefold after exercise compared to baseline (PRE) in all subjects but with different kinetics: almost 50% of the subjects had the highest miR-1 concentration in plasma just after exercise (POST), while the other half showed their maximal miR-1 expression 6 h after exercise (POST6). For all subjects, the miR-1 expression level decreased at POST12 and got closer to PRE-levels.

Interestingly, at POST, the miR-1 concentration was *very likely* and *likely* lower with P-CHO and CHO, respectively, compared to the PLA condition (*moderate*, ES: 1.0 and 0.7 for P-CHO and CHO, respectively). After 6 h and 12 h of rest, no further differences were shown between the three conditions.

#### 3.3.7. Inflammatory microRNA (miR-146a)

Plasma expression levels for miR-146a slightly increased at POST, with some interindividual differences. For each weekend session, about 30% of the subjects had a four- to fivefold increase for this microRNA, while 70% showed stable values compared to PRE. No differences were shown between the different conditions.

#### 3.3.8. Cardiac microRNA (miR-208a)

Cardiac miR-208a was not detectable in plasma samples throughout the study, indicating that a typical day of a rugby 7s tournament *likely* did not induce heart injury.

### 3.4. Muscle Soreness

No global effect of nutritional supplements was detected on the following markers of muscle soreness (Figure 4).

However, using a qualitative data analysis, if was shown that at 24 h post exercise, delayed-onset muscle soreness (DOMS) was *very likely* elevated compared to baseline with the P-CHO and CHO conditions (*moderate*, ES: 1) and *most likely* elevated with the PLA condition (*large*, ES: 1.2). At 48 h post exercise, DOMS remained *very likely* elevated compared to baseline with the three supplement conditions (*moderate*, ES: 1) (Figure 4A). At 72 h post exercise, DOMS returned to basal values for all conditions.

At 24 h post exercise, P-CHO *likely* reduced pain on muscle palpation compared to PLA (*small*, ES: 0.5). At 48 h post exercise, P-CHO and CHO *likely* decreased pain on muscle leg palpation compared to the PLA condition (*small*, ES: 0.5) (Figure 4B).

Moreover, the P-CHO condition *possibly* reduced muscle pain when descending stairs compared to the PLA condition at 24 h post exercise (*small*, ES: 0.4) (Figure 4C).

## 4. Discussion

Previous studies have shown that the intake of whey protein or carbohydrates after prolonged exercise may improve muscle protein synthesis, glycogen repletion and reduce muscle damage [10,11,12,13]. However, very few studies have investigated the effects of P-CHO on muscle performance and recovery following team-sport activities [14]. To the best of our knowledge, this is the first study that investigated the effect of different recovery drinks during an ecological situation of a rugby 7s tournament on performance and biomarkers of muscle damage and inflammation. We hypothesized that repeated rugby 7s matches would result in an increase in muscle damage and inflammation blood biomarkers, which in turn may have a negative impact on performance that would be lowered by a P-CHO supplement. Our results confirmed that native whey protein and carbohydrate supplementation after each rugby 7s match during a tournament day had a positive effect on performance, patterns of serum biomarkers of muscle damages and perception of muscle pain over a 24 h recovery period.

CMJ performance was not affected by the simulated tournament, nor by the different supplementation strategies, which is in line with the scientific literature in team sports protocols [28,29,30]. In the three supplement scenarios, sprint performance from 10 to 20 m was not altered at the end of the rugby tournament day. Previous studies have also shown no effect of proteins or carbohydrates supplementation on running sprint performance for a 15 to 30 m distance after a team sport match simulation [14,31,32]. However, interestingly, we showed that the time of a 50 m sprint was *likely* higher with the PLA condition compared to PRE (+1.7%, *moderate* increment), but was maintained with the P-CHO and CHO conditions. From a 50 m sprint, glycogen stores and glucose availability could potentially be a determinant factor in maintaining high-level performance [33], as the PLA condition significantly decreased performance at 50 m compared to PRE.

Compared to PLA, P-CHO ingestion *likely* helped to maintain running sprint performance after exercise, from 30 m (*moderate*) to 50 m (*large*). Interestingly, isoenergetic carbohydrate ingestion also *likely* helped to maintain sprint performance compared to PLA but with a lower effect (*small* at 30 and 40 m and *moderate* at 50 m). At 50 m, the P-CHO condition tended to help to maintain sprint performance compared to CHO (*small* positive effect). Our data suggest that the addition of native whey proteins to carbohydrate in recovery drinks may help maintain sprint performance from 30 to 50 m. Conversely, our results suggest that a 20 m sprint performance after a simulated rugby 7s tournament protocol would rely more on the ability to produce high levels of neural drive and the energy supply from the phosphagen system [33,34], as sprint performance was not altered after the match day and no difference were pointed out between the three conditions.

Similarly, we showed that endurance capacity at the end of the day was *likely* improved with P-CHO and CHO supplementation compared to PLA with a *small* effect. The addition of protein had no positive effect on endurance capacity, compared to the isoenergetic carbohydrate supplement. The benefit of adding protein to carbohydrate on glycogen repletion has been shown to be efficient when carbohydrate intake is suboptimal [3]. As expected, the addition of protein to a suboptimal amount of carbohydrate (60 g with the P-CHO condition) improved running endurance capacity to the same extent as in the carbohydrate optimal rate condition (CHO). The positive effect of P-CHO on sprint performance was significant from 30 m and seemed to be even more effective at 50 m. This may be the potential benefit of protein ingestion on muscle recovery.

A one-day rugby 7s simulated tournament induced significant muscle damage which was evidenced by a significant rise of CPK plasma concentrations and self-reported muscle soreness values. CPK concentration *most likely* increased after the tournament day and remained elevated after 12 h of rest, for all conditions. In parallel, general muscle soreness significantly increased after exercise and remained elevated for 48 h for the three conditions. Regarding the difference between beverage consumption, our results show that the rise of the CPK concentration, from baseline to POST6 was *likely* attenuated following the ingestion of P-CHO (+963.9 ± 837.4) compared to PLA (+1244.8 ± 617.2) (*moderate*). In the same way, P-CHO tended to *possibly* lower the CPK concentration compared to the CHO condition (65% chance to have a lower CPK concentration after P-CHO ingestion, and 7% chance with CHO ingestion, *small* effect, ES: 0.3). Despite this tendency, there was no significant difference between P-CHO and CHO nor between the the CHO and PLA conditions.

P-CHO and CHO *likely* and *very likely* lowered the increase of miR-1, a muscle-specific microRNA marker of muscle injury after the tournament day compared to the PLA condition (*moderate*). Even if our result suggested a better probability to reduce the increase of miR-1 with the P-CHO (*very likely*) compared to the CHO condition (*likely*), there was no significant difference between the two isoenergetic supplements. In accordance with the rise of the muscle damage biomarkers, the rugby 7s tournament induced an increase in muscle soreness for 48 h after the end of the exercise. At 24 h after exercise, leg muscle soreness when descending stairs was *possibly* lowered when the players were supplemented with P-CHO compared to PLA (*small*). Consistently, leg muscle pain at palpation was *likely* lowered with the P-CHO compared to the PLA condition (*small*) and tended to be lowered also compared to the CHO condition (54% chance to have lower muscle pain on palpation with the P-CHO beverage and 6% chance in favor of the CHO beverage compared to P-CHO, *small* effect). These results indicate that protein and carbohydrate supplementation during a rugby 7s tournament attenuates muscle damage and helps to maintain sprint performance from 30 m, compared to a placebo drink. Even if our result indicate a tendency in favor of a protein and carbohydrate supplement, the evidence is less clear when comparing it to an isoenergetic carbohydrate supplement, confirming the importance of carbohydrate availability during exercise on acute muscle and performance response to a high-intensity intermittent exercise [35]. During longer periods of supplementation, such as a one-week marathon preparation, Huang et al. (2017) showed that protein supplementation (33 g of whey protein) after each training session and after the marathon aided in attenuating the metabolism and muscular damage and the drop of performance compared to an isoenergetic carbohydrate beverage (33 g of maltodextrin) [36]. However, in the latter study, the quantity of carbohydrate for recovery was not optimal, and a daily nutritional assessment was not included. Therefore, calorie and daily macronutrient consumption may potentiate the effect of protein supplementation consumption on recovery and subsequent performance.

During a team sport match day competition, it is generally recommended to consume 5–10 g/kg of carbohydrates and 1.6–2 g/kg of proteins [37,38]. In the present study, the mean carbohydrate intake was 4.2 ± 1.1 g/kg in the PLA condition, and without carbohydrate supplements, the players did not meet the current carbohydrate recommendations for optimal performance. This was confirmed for all our results that concerned physical performance, muscle damage and inflammatory markers, which were optimized by additional carbohydrate intake. For protein intake, the rugby players were able to meet the recommended requirements with food on match day. This could explain the small effect of the P-CHO supplementation compared to CHO. The timing and type of proteins that were provided with the P-CHO condition (taking native whey proteins just after each exercise session) might have been too slight to be statistically significant compared to CHO. Another hypothesis would be related to the protein intake close to the recommendations for athletes with PLA and CHO supplements. This would be at the origin of a “muscle-full” effect [39]. This hypothesis can likely be rejected since the “muscle-full” concept relies on the *refractoriness* of muscle protein synthesis after a single intake, not on a daily protein intake. It is currently suggested that 0.4 g/kg/meal would optimally stimulate the muscle protein synthesis and this amount beyond which a muscle-full effect is expected is much higher than the protein intake at the end of each match with the P-CHO drink (20 g, ~0.27 g/kg) [40].

The present results suggest that a rugby 7s tournament induces a transient small change in TNF-alpha, IL1-ra and IL-6, inflammatory or anti-inflammatory biomarkers, mainly related to the eccentric workload [41]. Previous studies reported 2–2.5-fold increases in IL-6 immediately after intense running in trained athletes [42,43]. Compared with prolonged continuous aerobic exercise, in which 4 to 40-fold increases in the IL-6 concentration, the IL-6 response to high-intensity interval exercise appears to be lower [43]. Our study and previous studies confirmed that this type of exercise, with relatively brief overall exercise time, elicits a small inflammatory cytokine response [44,45,46,47]. Thus, the massive inflammatory response observed with prolonged continuous aerobic exercise [48,49], may be reduced if rest intervals are incorporated into intense bouts of exercise [46]. To our knowledge, this is the first study showing that a tournament day in rugby 7s induces an increase in IL-6 and IL1-ra concentrations, which remain elevated for 6 h after the last match. In the present study the supplementation with P-CHO *possibly* lowered the rise in IL-6 concentration after 12 h of recovery, compared to the CHO and PLA supplements (*small*). Previously, Kerasioti et al. (2013) also showed a decrease of IL-6 and CRP concentrations with the ingestion of a whey protein cake compared to an isoenergetic carbohydrate cake after a prolonged cycling exercise for 2 h [50]. This lowered IL-6 concentration with soluble milk proteins supplementation could be explained by the tryptophan content in the P-CHO drinks (0.8 g in each P-CHO drink, so a total of 2.4 g of tryptophan ingested during the tournament day). It is well established that melatonin concentration can lower inflammatory response [51,52], and the tryptophan via serotonin secretion is a precursor of melatonin. Surprisingly, 1 h after the last game of the day (POST), TNF-alpha concentration *most likely* decreased in all conditions and returned to baseline values after 12 h of rest with P-CHO and CHO, but not with the PLA condition. Previously, it has been shown that TNF-alpha increased during and immediately after exercise [44,51]. In our study, POST measurement was made 1 h after the last match, which might explain the decrease of TNF-alpha.

## 5. Conclusions

In conclusion, the present results suggested that a rugby 7s tournament was at the origin of muscle damage and inflammation in subelite players. Markers of muscle damage were potentially lowered by the ingestion of sport drinks containing either CHO only or P-CHO. These positive effects of beverages on inflammatory markers, and especially the protein carbohydrate beverage, allowed players to maintain higher levels of sprint and high-intensity performance at the end of the day. Consistent with previous studies, P-CHO supplementation could optimize recovery from repeated rugby 7s matches [11,15,44,51]. However, as this is the first study to show the impact of three repeated rugby 7s matches on muscle function and performance, further studies are needed to examine the effects on muscle damage and the maintenance of physical performance of different types of carbohydrates (i.e., low vs. high glycemic index) and amount of proteins in P-CHO beverages (i.e., 20 vs. 40 g).

### Limits of the Study

The study had some limitations. Firstly, the number of participants was small (12 subjects), and secondly, during the week, the subjects had to follow recommendations for training, sleep, alcohol consumption and nutrition. They had to self-report these daily by completing a questionnaire. However, the researchers relied on the integrity of the players to report these accurately as this was not objectively measured and quantified.

## 6. Practical Recommendations

From a practical standpoint, the results suggest that during a team sport tournament day, food may not be sufficient to match the optimal daily nutritional needs. During these competitive conditions, the addition of a sport-specific beverage providing 20 g of native whey proteins and 60 g of carbohydrates after each match is an effective strategy to optimize recovery [53].

## Figures and Tables

**Figure 1 nutrients-14-04780-f001:**
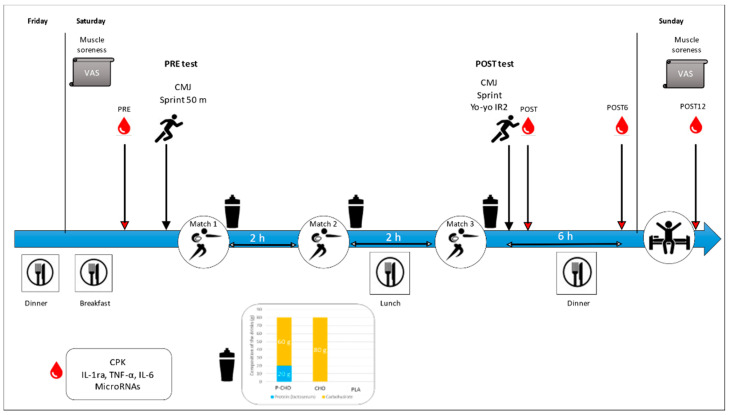
Experimental study design overview. Participants followed the same protocol for 3 consecutive weekends. The only difference was the drink consumed after each match: PLA, placebo drink; CHO, carbohydrate drink; P-CHO, protein and carbohydrate isoenergetic drink. PRE, pretournament tests; POST, post-tests at the end of the third simulated rugby 7s match; VAS: visual analog scale; CPK: creatine phosphokinase; IL-1ra, interleukin-1 receptor antagonist; TNF-α, tumor necrosis factor-alpha; IL-6, interleukin-6; CMJ, countermovement jump, Yo-Yo IR2, Yo-Yo intermittent recovery test 2.

**Figure 2 nutrients-14-04780-f002:**
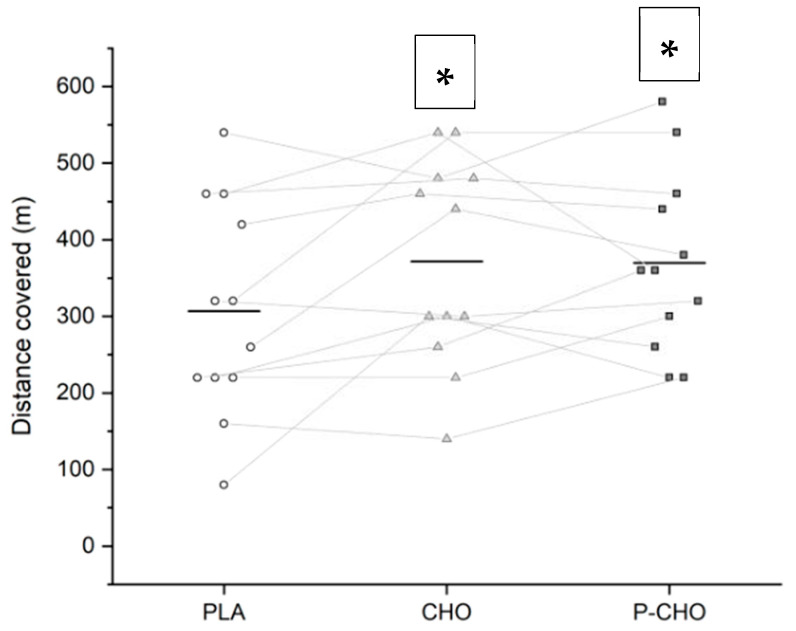
Distance covered during the high-intensity running performance test (Yo-Yo test) during PLA, CHO and P-CHO conditions. The line represents the mean. Individual distances are represented by white circles for PLA, grey triangles for CHO and black squares for P-CHO. * denotes a difference compared to PLA (*small*).

**Figure 3 nutrients-14-04780-f003:**
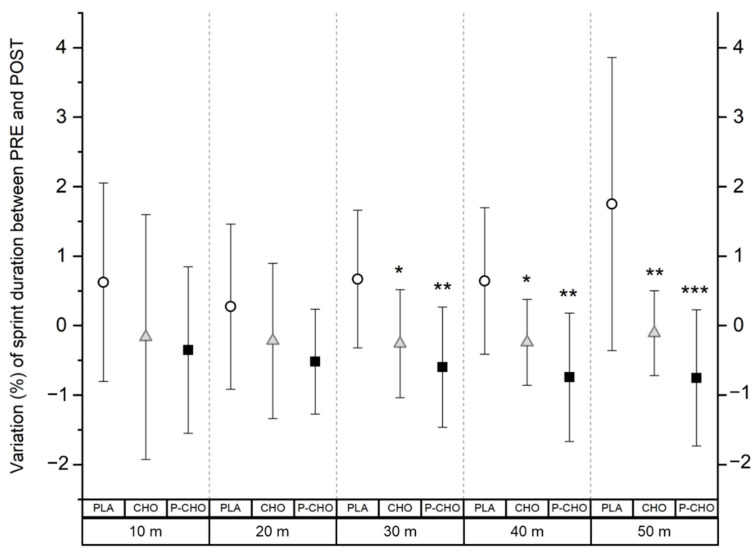
Change in sprint performance at the end of the tournament day compared to PRE (before the start of the first simulated match), from 10 m to 50 m distances for each condition. * denotes a difference compared to PLA (small). **, moderate difference compared to PLA. ***, large difference compared to PLA.

**Figure 4 nutrients-14-04780-f004:**
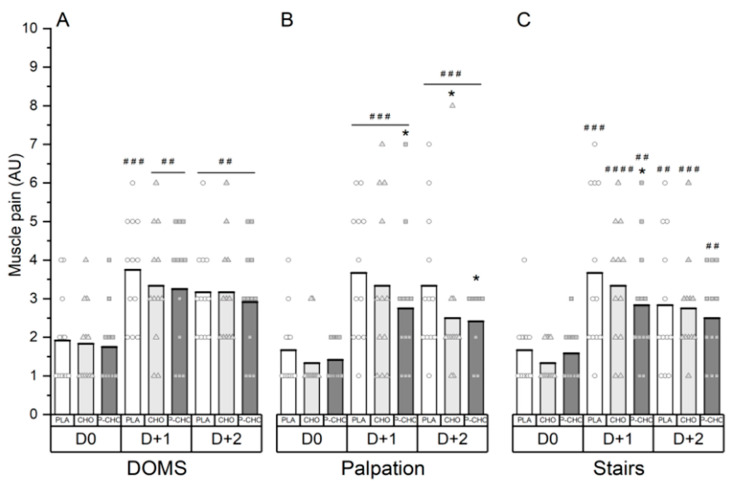
Muscle pain scores in general (**A**), at leg palpation (**B**) and when descending stairs (**C**). D0 for PRE situation; D + 1, 24 h after exercise and D + 2, 48 h after exercise. * denotes a difference compared to PLA (*small*). ^##^ denotes a difference compared to D0 (*moderate)*. ^###^, *large*. ^####^, *very large*.

**Table 1 nutrients-14-04780-t001:** Dietary intake during the rugby 7s tournament day.

	PLA		CHO		P-CHO		Statistic	
	Mean	SD	Mean	SD	Mean	SD	Quantitative	Qualitative
Macronutrients								
Energy (Kcal)	2656	559	3616 *	512	3616 *	512	*p <* 0.001	Most likely very large
Kcal/kg	32	7	43 *	8	43 *	8		
Protein (g)	157	32	152	23	207 *^µ^	23	*p <* 0.001	Most likely very large
g/kg	1.9	0.4	1.9	0.4	2.6 *^µ^	0.4		
% energy intake	23	-	17	-	23	-		
Carbohydrate (g)	359	104	564 *	80	509 *^£^	80	*p <* 0.001	Most likely very large ^£^ Most likely small
g/kg	4.2	1.1	7.1 *	1.2	6.3 *^£^	1.1		
% energy intake	52	-	64	-	58	-		
Lipid (g)	74	26	74	26	74	26	*p =* 1	
g/kg	0.9	0.3	0.9	0.3	0.9	0.3		
% energy intake	25	-	19	-	19	-		
Micronutrients								
Vitamin C (mg)	141	99	141	99	141	99	*p =* 1	
Magnesium (mg)	533	77	533	77	533	77	*p =* 1	
Sodium (mg)	3360	614	3360	614	3360	614	*p =* 1	
Potassium (mg)	4378	1142	4378	1142	4378	1142	*p =* 1	
Iron (mg)	22	4	22	4	22	4	*p =* 1	

The quantitative statistical analysis was performed using the Friedman test. *, different from PLA; ^µ^, different from CHO; ^£^, different from CHO with qualitative statistic. PLA, placebo condition; CHO, carbohydrate condition; P-CHO, protein and carbohydrate isoenergetic condition.

**Table 2 nutrients-14-04780-t002:** Markers of muscle damage and inflammation.

	PRE		POST		POST6		POST12	
	Mean	SD	Mean	SD	Mean	SD	Mean	SD
CPK (IU·L^−1^)								
PLA	271	184	1043 ^#^ *Large*	522	1245 ^#^ *Large*	617	1010 ^#^ *Large*	554
CHO	253	114	915 ^#^ *Very large*	757	1011 ^#^ *Very large*	574	810 ^#^ *Very large*	516
P-CHO	165	95	1073 ^#^ *Very large*	1239	964 ^#^* ^*#*^ *Very large * * *Moderate*	837	1000 ^#^ *Very large*	865
TNF-α (pg·mL^−1^)								
PLA	0.97	0.1	0.78 ^#^ *Large*	0.1	0.83 ^#^ *Large*	0.1	1.01 ^#^ *Small*	0.1
CHO	0.99	0.1	0.79 ^#^ *Large*	0.1	0.86 ^#^ *Moderate*	0.1	1.01	0.1
P-CHO	0.97	0.1	0.78 ^#^ *Large*	0.1	0.82 ^#^ *Large*	0.1	0.97	0.1
IL-1ra (pg·mL^−1^)								
PLA	218.9	69.3	450.1 ^#^ *Large*	187.1	386.2 ^#^ *Large*	120.2	260.2 ^#^ *Small*	93.0
CHO	210.7	64.7	391.8 ^#^ *Large*	131.7	334.2 ^#^* *^#^ Large * * *Small*	103.7	220.9 * * *Small*	66.6
P-CHO	210.0	60.2	419.4 ^#^ *Very large*	189.9	332.0 ^#^* *^#^ Large * * *Small*	116.1	208.8 * * *Small*	56.6
IL-6 (pg·mL^−1^)								
PLA	4.7	8.2	10.8 ^#^ *Moderate*	6.9	5.9 ^#^ *Small*	8.0	5.7 ^#^ *Small*	7.9
CHO	4.0	7.3	8.9 ^#^ *Moderate*	6.9	5.5 ^#^ *Small*	7.2	6.0 ^#^ *Small*	8.9
P-CHO	3.5	6.7	9.2 ^#^ *Moderate*	6.7	5.0 ^#^ *Small*	6.5	4.1 *^£ ^ * *Small* *^£^ Small*	6.7

^#^ Different from PRE. * Different from PLA at the same time. ^£^ Different from CHO at the same time. CPK: creatine phosphokinase; TNF-α: tumor necrosis factor-alpha; IL-1ra: interleukin 1 receptor antagonist; IL-6: interleukin 6. PLA, placebo beverage; CHO, carbohydrate beverage; P-CHO, protein and carbohydrate isoenergetic beverage.

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
