# Peer review of "Effects of Native Whey Protein and Carbohydrate Supplement on Physical Performance and Plasma Markers of Muscle Damage and Inflammation during a Simulated Rugby Sevens Tournament: A Double-Blind, Randomized, Placebo-Controlled, Crossover Study"

_nutrients, 2022, doi:10.3390/nu14224780_

Round 1
Reviewer 1 Report
Dear authors,
your study addresses an interesting topic, the optimal nutrition strategy to alleviate muscle damage and maintain performance during contact team sports tournaments. The use of a simulation is a good idea as it ensures repeatability and control. Unfortunately, I found the frequent errors in English a bit distracting, if these are improved then the quality of the manuscript will be greatly enhanced. I have a few specific and general comments for you.
Specific comments:
Abstract
The abstract does not fully represent what was actually done in the study. It is not clear that three simulated tournaments, over three weeks, were performed. Currently it appears that three simulated games were performed and a different drink was given after each game. Please change the abstract to reflect the design of the study.
Define CPK
Introduction
The introduction is appropriate and provides relevant background information.
Methods
Why is it deemed a pilot study?
I commend you for the control you had in place for the diet, this appears to have worked well and ensured a difference in carb, protein and energy between the treatments. Was a diet dairy used in the days prior to ensure carb intake, in particular was the same each week?
Why was peak sprint performance not reported? You suggest that this was done to improve reliability however was there much difference between the two results - one poor performance would increase the average down and could impact your analysis.
Statistical analysis and results
Given the repeated bout effect, which may influence the response to exercise that involves eccentric contractions (decelerations and changes of direction, for example), it may be relevant to carry out statistical analysis to identify whether there is an order effect? Ignoring the drinks consumed, was there a difference in all measures between the three simulated tournaments? For example, was there less muscle damage in the second and third tournaments, compared to the first? This could have implications for your results.
GPS was used to measure training load, can this data be included in the results to show the simulations had the same physiological demands? Otherwise there appears to be no reason to mention the use of GPS.
Similarly, where are the results for the Hooper questionnaire?
Discussion
Protein intake was near the limit recommended for athletes and therefore the addition of more may not necessarily have any additional benefit (muscle full effect).You elude to this on line 583, however you could provide a reference to support this concept.
On line 528 you mention the correlation between CPK and muscle soreness, can you run a correlation to see if this indeed the case for your results? The relationship between CPK and soreness is not always clear cut, with soreness usually peaking at 48h and CPK peaking later, often at 72h post exercise. You have only captured the early responses at which time both will be slowly increasing.
The use of the magnitude-based inference terms (moderate, small, possible etc) could be better in the discussion. They do not flow well and feel out of place in some areas. The section line 588-621 is a good example of how they could be used, this is well written and flows nicely.
General comments:
I suggest using the term "soluble milk protein", instead of "milk soluble protein" throughout.
Be consistent with the use of protein-carbohydrate and P-CHO. You could use P-CHO throughout instead of "the mix protein-carbohydrate"
Reviewer 2 Report
The authors submitted a manuscript entitled ‘’Effects of soluble milk protein and carbohydrate supplement 2 on physical performance and plasma markers of muscle damages and inflammation during a simulated rugby seven tournament: a double-blind, randomized, placebo-controlled, cross-over study’’ investigating the effects of a CHO and P-CHO supplement on performance, muscle damage and inflammation during a congested rugby fixture. The study has potential significance and could add novel insights in the field of sports nutrition. However, there are several methodological flaws that hamper the quality of the research and make the proposed results difficult to follow.
More specifically:
· From a methodological point of view, the 3 trials should have been iso-energetic. In the PLA trial, the authors used a non-energetic placebo drink. Consequently, the 3 trials differed regarding total energy intake. Lower energy consumption in the PLA trial may have negatively affected performance as energy intake and muscle glycogen content (which is a result of CHO ingestion) are closely related to high-intensity actions in intermittent-type sports. Thus, differences between trials may be attributed to the different energy intake of the participants and not the supplement tested.
· Diet control before and during the study was sub-optimal. Participants' dietary profile was not monitored and adjusted before the study and as a result, potential differences in players’ protein intake (g/kg/day) may have influenced the results, especially during the 1st trial. Moreover, antioxidant intake was not considered nor discussed throughout the study. Antioxidants have been shown to affect exercise-induced muscle damage and inflammation post-exercise via the attenuation of oxidative stress and ROS production. Consequently, differences in the antioxidant intake may have influenced muscle damage and inflammatory responses. Prior and during the study protein and antioxidant intake should have been monitored and controlled to allow for safe conclusions regarding the supplements tested. Additionally, nutritional data are not reported properly. A table with macro- and micro-nutrient intake monitor and control should have been presented.
· Another issue is the statistical analysis used. I appreciate the authors’ attempt to present the effect sizes but this approach is by its nature qualitative and does not provide quantitative results. Parametric or non-parametric (based on normality check) tests should have been used and exact p values reported along with effect sizes. Based on this, the results in this form are not acceptable.
· The study is a clinical trial in humans. However, it has not been registered in a public clinical trial registry (i.e., clinicalTrials.gov).
· The manuscript needs extensive editing of the English language and style from a native English speaker.
